# The structure of the mammalian bornavirus polymerase complex

Loïc Carrique [1,4], Franziska Günl [2,4], Adrian Deng[3], Jonathan M. Grimes[1] & Jeremy R. Keown [1,3] ✉

Borna disease virus 1 (BoDV-1) is a non-segmented RNA virus with one of the smallest known RNA virus genomes. BoDV-1 replicates in the nucleus of infected cells using a virally encoded polymerase complex composed of the large protein and phosphoprotein. Here, we present the BoDV-1 polymerase complex at resolutions up to 2.8 Å, describing the fully ordered large polymerase protein bound to tetrameric phosphoprotein. The complex is maintained through the ordered C-terminal region of one copy of the phosphoprotein. Analysis of the model reveals a conserved methyltransferase domain, though key S-adenosyl methionine binding residues are missing. While no RNA is observed in our models, analysis of a sample under reaction conditions induces an opening and closing of the template entry and exit channels, respectively. Higher-order polymerase assemblies suggest oligomerisation as a conserved feature of negative strand RNA virus polymerases. We provide a molecular framework to investigate bornavirus replication and transcription.

Bornaviruses (family Bornaviridae) are a family of viruses that contain a non-segmented negative-sense RNA genome, which infect a diverse range of animals, including avian, reptilian, and mammalian species[1,2]. Some species of bornavirus, including Borna disease virus-1 (BoDV-1) and variegated squirrel bornavirus 1, can cause fatal infection in humans[3,4]. Presently, human infection with bornaviruses appears localised to Western Europe[5]. Unlike most other members of the *Mononegavirales* order, bornaviruses perform transcription and replication in the nucleus of infected cells[2]. Bornavirus has one of the smallest genomes among non-segmented negative-sense RNA viruses (nsNSV) at 8.9 kilobases and encodes at least six proteins[2]. The single strand of the viral genome is packaged into ribonucleoprotein (RNP) complexes by many copies of the viral nucleoprotein (*N*-protein), which protect and shield the genome from degradation in the infected cell. This RNP assembly provides a substrate to which the viral large protein (L-protein) is recruited.

The L-protein is a 192 kDa RNA-dependent RNA polymerase (RdRp) which forms the core of the replication complex. The L-protein performs transcription and replication, producing viral mRNA and new copies of the viral genome, respectively. To efficiently transcribe and replicate the genome, the L-protein associates with the viral phosphoprotein (P-protein). The P-protein is a 23 kDa protein that forms a tetramer via a helical oligomerisation domain and is flanked by N- and C- termini which are unstructured in solution[6,7]. The P-protein has multiple functions during the viral lifecycle, including linking the L-protein to RNA-bound N-protein during replication and transcription, and recruiting RNA-free N-protein to the site of replication. The complex of L-protein and P-protein is termed the replication complex.

Commonly, L-proteins from the *Mononegavirales* order contain three distinct functional domains. The RdRp domain catalyses nucleotide polymerisation templated by the viral RNA, whilst the polyribonucleotidyltransferase domain (PRNTase) adds the guanosine cap to the 5′ end of the nascent RNA. The methyltransferase domain (MTase) performs both 2′-O methylation and 7-N methylation producing a cap-1 structure. In addition, there are two non-catalytic domains, the connector domain (CD) and the C-terminal domain

[1]Division of Structural Biology, Centre for Human Genetics, University of Oxford, Oxford, UK. [2]Sir William Dunn School of Pathology, University of Oxford, Oxford, UK. [3]School of Life Sciences, University of Warwick, Coventry, UK. [4]These authors contributed equally: Loïc Carrique, Franziska Günl. ✉e-mail: jeremy.keown@warwick.ac.uk

(CTD), which have a largely unknown function. The structures of the L-proteins of the *Mononegavirales* have been reviewed in refs. 8–10.

Driven by developments in cryo-electron microscopy (cryoEM) and protein expression, the structures of several nsNSV L-proteins and replication complexes have been determined for species of Paramyxoviruses[11–15], Filoviruses[16,17], Rhabdoviruses[18–20], and Pneumoviruses[21–24]. These structures have an overall conserved architecture with a large N-terminal RdRp of approximately 900 amino acids, which packs tightly against the PRNTase domain. In a subset of structures, the CTD (MTase, CD, and CTD) are sufficiently ordered to be visualised[12,13,15,18,19,25] and observed in several conformations[9]. The L-P interface has been characterised for Paramyxoviruses, Filoviruses, and Pneumoviruses and involves extensive interactions from extensions at the C-terminal end of the tetrameric P-protein with the RdRp of the L-protein[9]. L-P structures from the Rhabdoviruses family describe an interaction via an N-terminal region of P-protein which winds around the L-protein CTD[18,19]. In contrast to the members of the *Mononegavirales* order above, bornaviruses contain a considerably smaller L-protein (192 vs ~250 kDa) and P-protein (23 kDa vs 30–80 kDa).

Here we present several structures of the BoDV-1 replication complex determined to global resolutions of up to 2.8 Å by cryoEM. One of these structures describes the full-length L-protein in complex with a tetramer of the viral P-protein. Collectively, analysis of these structures reveals the mode of L-P binding, the location of an inactive MTase domain, and dimeric L-protein complexes, broadening our understanding of the *Mononegavirales* order.

## Results

### Determining the structure of the BoDV-1 replication complex

To generate a complete stable L-P complex, we constructed a single vector encoding full-length genes of the BoDV-1 L- and P-proteins. SDS-PAGE confirmed the purification of a complex containing both proteins and the composition of the complex was determined in solution using mass photometry (Fig. 1a, b). Sample 1 was subsequently analysed by single particle cryoEM (Supplementary Fig. 1a) and diverse high quality 2D classes were generated (Supplementary Fig. 1b). While most particles were monomeric a small fraction of protein complexes in a tetrameric arrangement was observed (Supplementary Fig. 1c). However, only one orientation of the tetrameric complex was observed, preventing an accurate 3D-reconstruction. Consensus refinement of the monomeric particles yielded a map to a global resolution of 3.13 Å with further classification revealing two volumes. Ninety-seven percent of picked particles were reconstructed to a global resolution of 3.07 Å describing a high-resolution view of the RdRp core with local resolutions near 2.8 Å and lower resolutions for the CTD and P-protein regions.

An initial molecular model was generated using Alphafold[26] and alongside previously determined nsNSV L-protein structures were used to guide the building of a near full-length L-protein in complex with C-terminal regions of the P-protein (Fig. 1c–f). The N-terminal 19 residues, two internal loops 1061-1082 and 1124-1144, and an interdomain linker 1241-1253 could not be observed in the final L-protein model. In addition to the L-protein, a tetrameric assembly of the P-protein was observed in the complex with residues 118–166 observed in three of the chainsC/D/E and residues 118–182 observed for chainB. In addition to the protein components two ions, likely zinc, based on the tetrahedral coordination, were observed in the model[27]. Site 1 was coordinated by C993, E1020, C1210, C1213 and site 2 was coordinated by residues H1485, D1509, C1511, and C1586 (Fig. 1g). The ion in site 1 is similar to that previously observed in the canine parainfluenza virus-5 (PIV-5)[11], Rabies virus (RABV)[18], and Vesicular stomatitis Indiana virus (VSIV)[19] structures. This appears to be structurally, not catalytically, important given its relative location to the

PRNTase catalytic residues. We were unable to identify equivalent zinc ions in site two among other nsNSV L-protein structures.

The L-protein is formed by five domains, and at least two contain enzymatic activity. The N-terminal 797 residues of BoDV-1 L-protein forms the RdRp domain, which is smaller than the ~900 residue RdRp from other nsNSV (Fig. 1c, d). Annotation of the RdRp describes the location of the canonical N-terminal, thumb, fingers, and palm sub-domains (Fig. 2a). Alternative annotation can be used to show the position of motif's A-E from the palm subdomain and motif F/G from the fingers subdomain (Fig. 2b). These conserved motifs are involved in nucleotide and RNA binding during replication and transcription. The RdRp active site is formed by the conserved 656-GDNQ-659 motif, which is at the tip of the β-strand formed by motif D. D657 is expected to coordinate the conserved magnesium in the RdRp active site. Comparison of the Ebola virus L-protein bound to RNA, suggests template RNA would follow a similar path in BoDV-1 (Fig. 2b).

To assess the enzymatic activity of the purified BoDV-1 L-P complex, we performed in vitro RNA synthesis assays using 13- and 14-nucleotide (nt) templates derived from the leader sequence (Fig. 3). The templates produced similar products with a strong signal at 6 nt and the weakest at 10 nt. From the pattern of radionucleotide incorporation, we propose that these reactions have initiated internally on the template at positions 4 and 5 of the 13- and 14-nt templates, respectively. Previous work has shown that BoDV vRNA and cRNA promoters contain essential nontemplated extensions at the 3′ termini, and that these are required for initiating RNA synthesis[28]. Our results are consistent with these previous observations, supporting that initiation on these templates occurs after the 5′-AAC-3′ or 5′-AACA-3′ sequence at the 3′ termini, respectively. These extensions were not copied in our assays, suggesting their role may be important in positioning the template. The underlying mechanism by which the L-P complex restricts RNA synthesis to internal initiation or truncated products remains unclear.

### Organisation of the PRNTase domain

The PRNTase domain of nsRNA L-proteins catalyses the addition of a GTP to form the 5′ cap to nascent viral mRNA. The catalytic site of the PRNTase domain is formed from two highly conserved motifs found in the intrusion loop and the priming loop. The intrusion loop contains the HR-motif with the histidine residue that is covalently linked to the nascent RNA product during the capping reaction. The priming loop contains the GxxT motif, which is involved in binding the capping guanosine nucleotide. In BoDV-1, sequence and structural alignments identify 1135-HR-1136 and 1073-GEIT-1076 as the HR and GxxT motifs, respectively. Neither the priming loop nor the intrusion loop is ordered in our models. This is in contrast to several other nsNSV L-protein structures, where these loops have been observed in several arrangements linked to functional states[15]. These mobile loops were also not observed in recent Ebola virus and Nipah virus L-protein structures in the absence of RNA. In a later Ebola virus model, these became ordered upon addition of RNA[16,17,29]. Within the *Bornaviridae*, the BoDV-1 RdRp and PRNTase sequences appear well conserved, while the C-terminal regions show a much greater sequence diversity (Supplementary Fig. 2)

To capture snapshots of the complex during catalysis, we prepared a reaction mixture containing BoDV-1 L-P complex, a 23mer template RNA, a 6mer triphosphorylated primer, S-Adenosyl methionine (SAM), and a nucleotide triphosphate mixture (Supplementary Fig. 3a). A large single-particle cryoEM dataset was collected on sample 2 (Supplementary Fig. 3 a,b). From this sample, two related structures could be reconstructed, which differ only by the presence of a weakly ordered C-terminal region. The structure that did not contain the C-terminal region was used for further analysis, as it is of higher resolution in the regions of interest, and accordingly, we can be more confident in the domain movements. A third structure comprising only

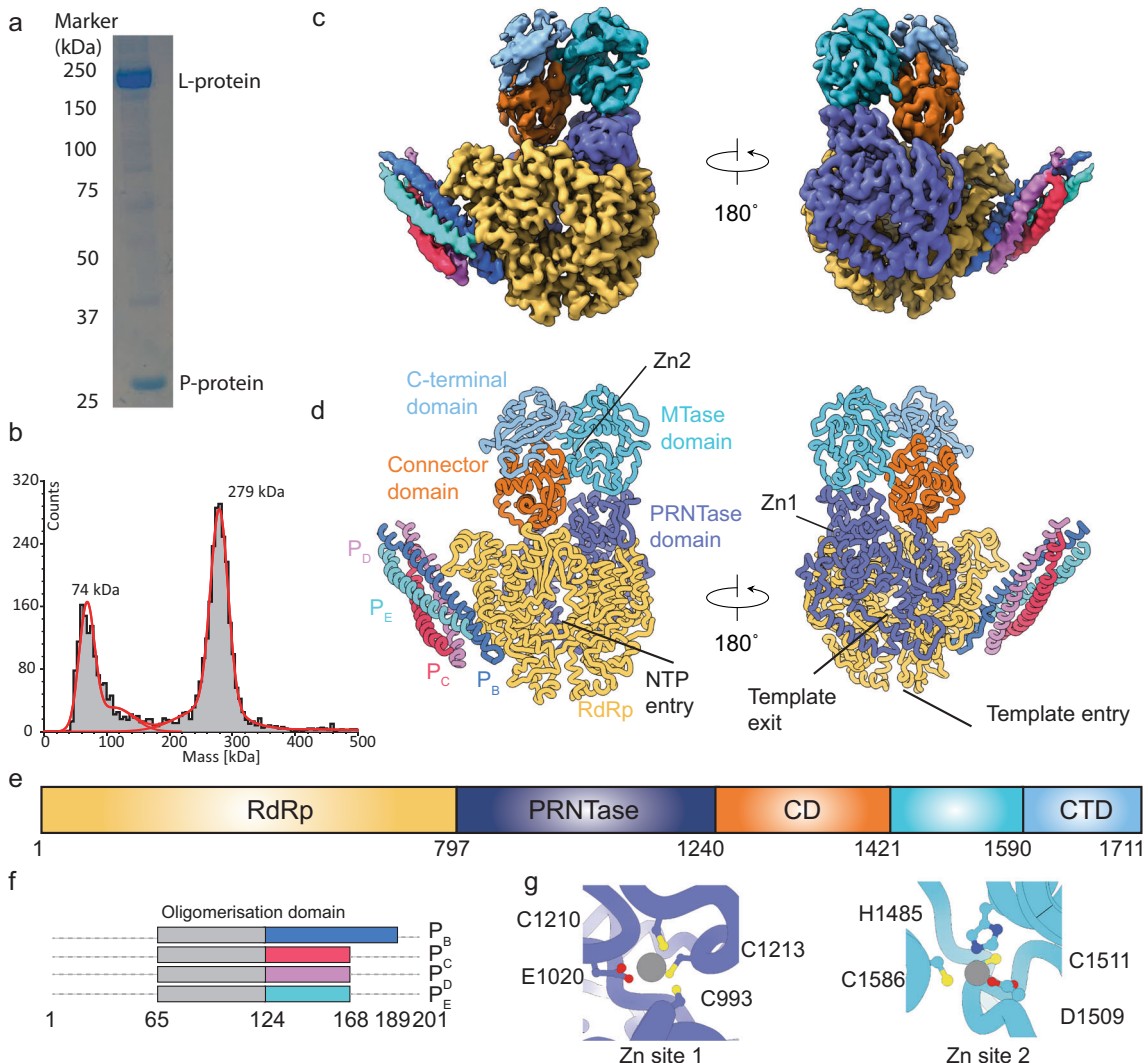

**Fig. 1 | Structure of the RNA free BoDV-1 L-P complex. a** SDS-PAGE of the sample used for analysis by cryoEM. The SDS-PAGE was run once for this sample. Source data are provided as a source data file. **b** Mass photometry data showing the presence of two major species in solution. Grey bars show number of counts and red lines the data fits. The L-P tetrameric complex has an expected molecular weight of ~280 kDa while the P-protein tetramer is expected to be ~86 kDa. **c,d** Unsharpened cryoEM map and model coloured by domain identity. **e,f** Functional regions of the L-protein (e) and P-protein (P) observed in our model with annotations. Colouring in these figures matches that of the figures in panel (**c,d**). The RdRp/PRNTase (grey), connector domain (orange), MTase domain (cyan), and C-terminal domain (blue) are shown for the L-protein. The four phosphoprotein molecules are shown in dark blue, pink, purple, and dark green). **g** coordination of the two zinc (Zn) ions (represented as grey spheres) in our model with the coordinating residues annotated. The protein backbone is coloured as in figure c-e, sulphur atoms (yellow), nitrogen atoms (blue), and oxygen atoms (red).

the RdRp and PRNTase domains was present, which was identical to the structure determined from sample 1.

While no RNA, NTP or SAM was observed in these structures, comparison of the RNA-free and reaction structures revealed significant rearrangements in the PRNTase domain and movements in the RdRp core (Fig. 2c, d). In sample 2, residues 727–735 from the RdRp have become disordered and residues 400–407 have retracted away from the active site (Fig. 2e). The changes were larger in the PRNTase domain where two helices (residues 875–905) have moved by approximately 7 Å. The adjacent helix formed by residues 947–956 has moved by 8 Å and residues 957–960 have become disordered (Fig. 2c, d). Residues 1168–1177 have also become disordered. Collectively, these movements in the PRNTase domain and RdRp have caused an expansion of the template entry channel such that a modelled template from the recent Ebola virus L-protein structural model could be accommodated without clashes and a tightening of the template exit channel. While these movements would suggest the presence of RNA in the L-protein, we were unable

to assign any density to RNA template or primer. During manuscript review an elongation complex was determined for the Nipah virus L-P complex[30], this duplex RNA is more easily accommodated in our sample 2 model though also not without clashes suggesting further rearrangements are required for RNA extension. Our attempts to capture RNA-bound complexes, using the 14mer or 23mer RNA at 10X molar concentrations to BoDV-1 L-protein, in the presence of the 6mer primer, or with NTP at low sodium chloride concentrations, were unsuccessful.

## Analysis of the C-terminal region of BoDV-1 L-protein

From the two datasets we collected, maps could be reconstructed which contained the full-length L-protein, including the flexible C-terminal region. The C-terminal regions arranged similarly in the models, and we used the RNA free model to describe the positions, as the domains are at higher resolution.

As expected from the smaller overall L-protein size of BoDV-1, the individual domains in the C-terminal region are significantly smaller

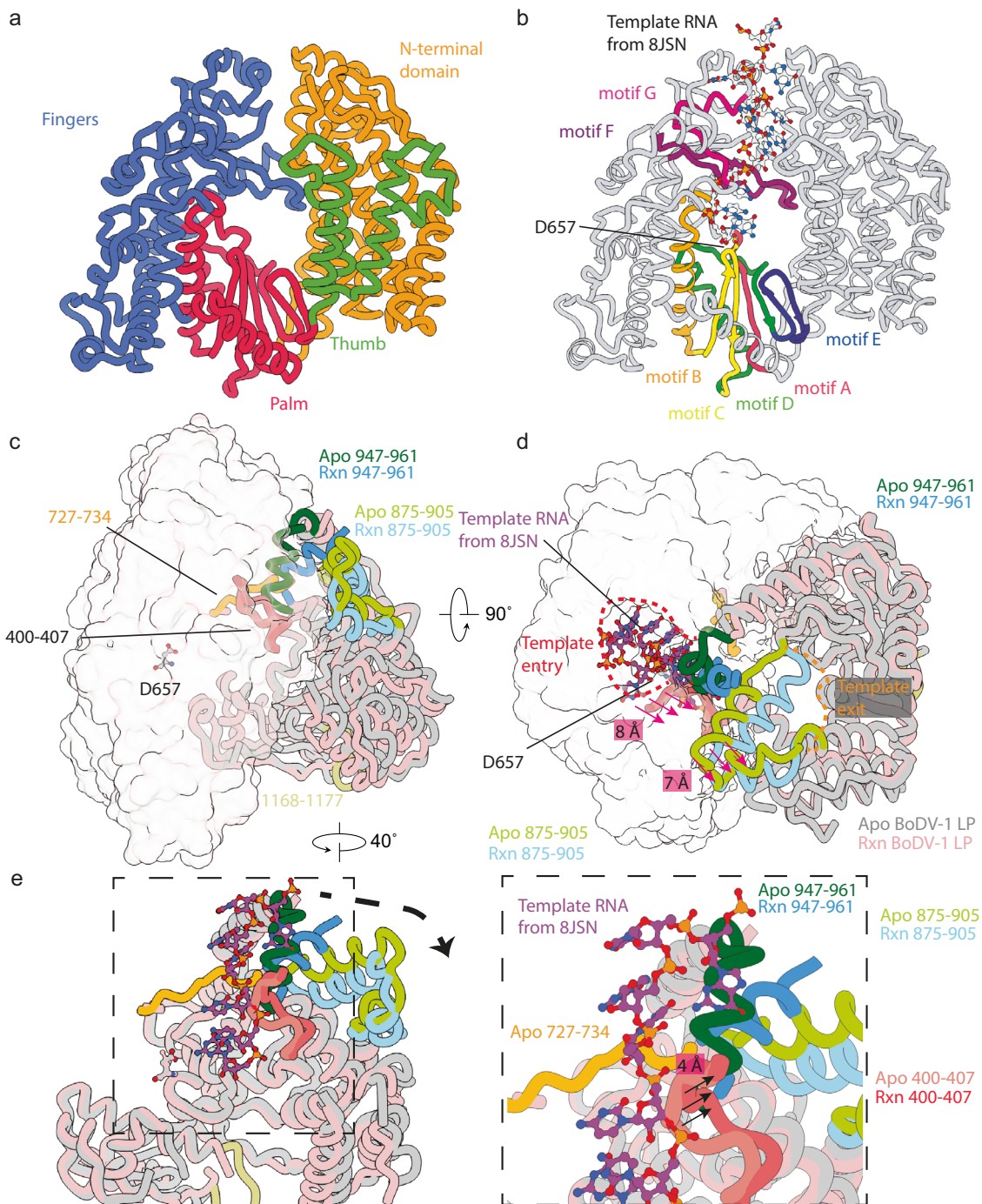

**Fig. 2 | Details of the L-protein RdRp and PRNTase domains. a** Annotations of the RdRp core showing the N-terminal domain (orange) and fingers (blue), palm (red), and thumb (green) subdomains. **b** Annotations for the RNA and nucleotide binding motif A (red), motif B (orange), motif C (yellow), motif D (green), motif E (blue), motif F (purple), and motif G (pink). The remaining RdRp domain is shown in grey. The RdRp active site D657 is show as stick representation. In **b** template from the Ebola RNA polymerase structure (PDB ID 8JSN, light blue) is shown demonstrating the path of the 3' RNA template into the RdRp active site. **a** and **b** are shown in identical orientations with residues outside the RdRp hidden for clarity. **c,d** movements of residues from the PRNTase and RdRp under reaction conditions. Regions of the RdRp core which are unchanged are shown in transparent surface representation. Regions of the PRNTase and RdRp which undergo significant movement are annotated as are the approximate locations of the template entry and exit channels. The view in panel (**d**) is rotated 90° from (**c**). In (**d,e**) template from the Ebola RNA polymerase structure (PDB ID 8JSN, purple) is shown demonstrating the path of the 3' RNA template into the RdRp active site. **e** rotation and zoom showing the movement of the 400-407 and 727-734 region of residue in the Apo and Rxn models. The template RNA from PDB ID 8JSN is used to demonstrate a clash with the residues 947-961 in the Apo but not in the Rxn model. In panels (**c–e**) the Apo RdRp (shades of green and grey) and RxN samples (shades of blue and pink) are shown.

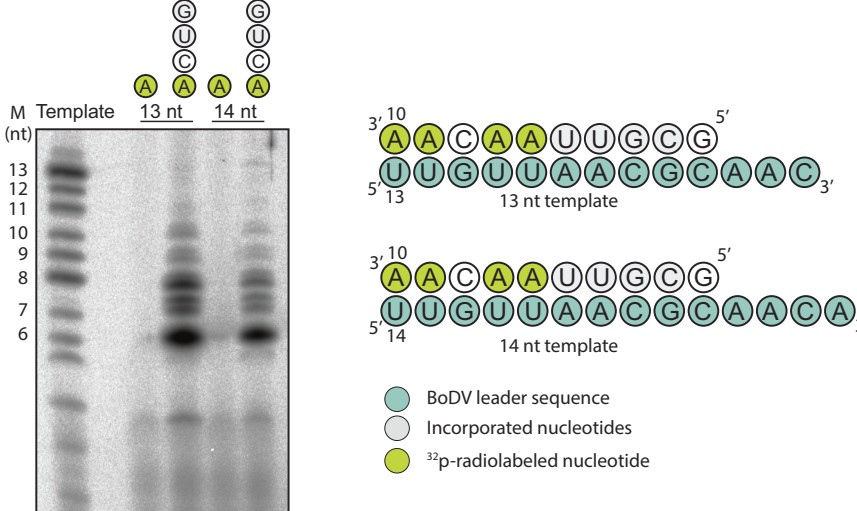

**Fig. 3 | In vitro incorporation assay.** In vitro [α³²P]-ATP incorporation assays for the 3′ extension activity of the L-P complex assessing elongation on a 13 or 14 nt template. Assays were performed in the presence of only [α³²P]-ATP or [α³²P]-ATP with CTP, UTP, and GTP. A radio labelled 13-mer RNA sequence was used as a marker (M) with nt lengths indicated on the left. Source data are provided as a source data file. Template nucleotides (green), incorporated nucleotides (light grey), and ³²p ATP nucleotides (yellow) are shown. The experiment was repeated three times with the same result.

than those of other nsNSV L-proteins (Supplementary Fig. 4). Structural alignment showed some homology to the CD from hPIV3 (RMSD = 1 Å across 45 atom pairs) but no homology to other determined nsNSV L-protein models (Supplementary Fig. 4). The CD were between 90 and 140 residues larger in the other nsNSV L-proteins than in BoDV-1 (Supplementary Fig. 4). Alignment of the BoDV-1 CTD domain showed weak structural homology to those from other nsNSV. A structure-based phylogenetic analysis was performed using the Foldmason server V1[31] using representatives of the experimentally determined L-protein structural models (Supplementary Fig. 5). This phylogenetic tree clustered the seven *Paramyxoviridae* and two *Rhabdoviridae* structures together, with the *Bornaviridae* closest to the root of the tree.

In nsNSV L-proteins the MTase domain is located between the CD and CTD, where it catalyses methylation of the nascent RNA cap during transcription. Canonical MTase activity requires the conserved catalytic K-D-K-E tetrad of residues and the GxGxG motif for binding of the methyl donor SAM[10]. An sequence analysis of the BoDV-1 C-terminal region does not reveal residues that would be consistent with these motifs. In our model, the MTase (residues 1421–1590) forms a tight globular domain centred around two antiparallel α-helices, which are surrounded by a large β-sheet (Fig. 4a, b). A structural homologue search using the Foldseek webserver[32] gave a large number of hits, many of which were annotated as MTase domains. The top hit from the Protein Data Bank was a MTase from *Mesembryanthemum crystallinum*, a desert plant. The overlay of these two proteins showed good homology (RMSD = 1.06 Å across 20 atom pairs) to a core region of the plant MTase from residues 56–121 (Fig. 4b). The plant MTase contains a S-Adenosyl-L-homocysteine molecule (SAH). Overlay with the BoDV-1 MTase shows the SAH (or SAM) would not be accommodated without significant structural rearrangement of the MTase. The absence of important motifs and the structural restrictions of the domain to accommodate substrate, suggests BoDV-1 L-protein does not contain an active MTase. We currently do not have a well-supported hypothesis for how or if BoDV-1 mRNA transcripts would be methylated in vivo.

### Analysis of the P-L interaction interface

In the BoDV-1 replication complex, one copy of the L-protein was found in complex with a tetramer of the P-protein. The oligomerisation domain of the tetramer is identical to the X-ray crystallography structures determined recently[6,7]. The interaction of the P-protein with L-protein buries a surface area of ~1440 Å². The interaction surface on the L-protein is completely contained within the RdRp domain, while from the P-protein chainB contributes 1324 Å² of the surface and chainC contributes the remaining 116 Å². Approximately half of the chainB interaction interface area is from residues along one face of the α-helical oligomerisation domain from residues 143–165 (Fig. 4c, d). The interaction between chainB and the RdRp is maintained by strong electrostatic interactions with the negatively charged helical interface interacting with a complimentary electropositive interface on the L-protein. The electropositive interface on the RdRp comprises residues R568, R581, and H352, while E150 and E157 on the P-protein provide the opposing negative charges (Fig. 4d).

Residues C-terminal to the oligomerisation domain (168–182) of chainB have become ordered and pack between chainD of the P-protein tetramer and the RdRp. This interface buries approximately 650 Å² and is maintained by a large hydrophobic surface. This stretch of fifteen residues contains five proline residues. In the recently determined BoDV-1 P-protein crystal structures these residues were not observed[6,7]. Chains D and E are maintained in the complex through tight association in the P-protein tetramer.

The interaction sites of the P-protein with other viral proteins have previously been investigated. The four C-terminal residues have been reported to be absolutely required for interaction with bornavirus N-protein[33]. The L-protein binding site on P-protein was localised to the C-terminal region. In a luciferase reporter assay, a construct comprising P-protein residues 1–172 did not support L-protein activity, while extension to residue 183 recovered most activity[34]. Comparison to our structure (Fig. 4d) shows that the critical residues 172–183 are those that in chain B of our model become ordered and form approximately half of the interaction area with L-protein, validating the biological importance of this interface for viral function. These studies also mapped the overlapping interaction interfaces of the P-protein to interact with L-protein and other copies of P-protein[33,34]. Here we can explain this as residues 135–172 for the oligomerisation domain helix, with one face binding other copies of P-protein and the other forming a large interface with the L-protein RdRp domain (Fig. 4d).

This interaction of the tetrameric P-protein with the RdRp of the L-protein is at an approximately equivalent location to that observed in recent filovirus, pneumovirus, and paramyxovirus L-P structures,

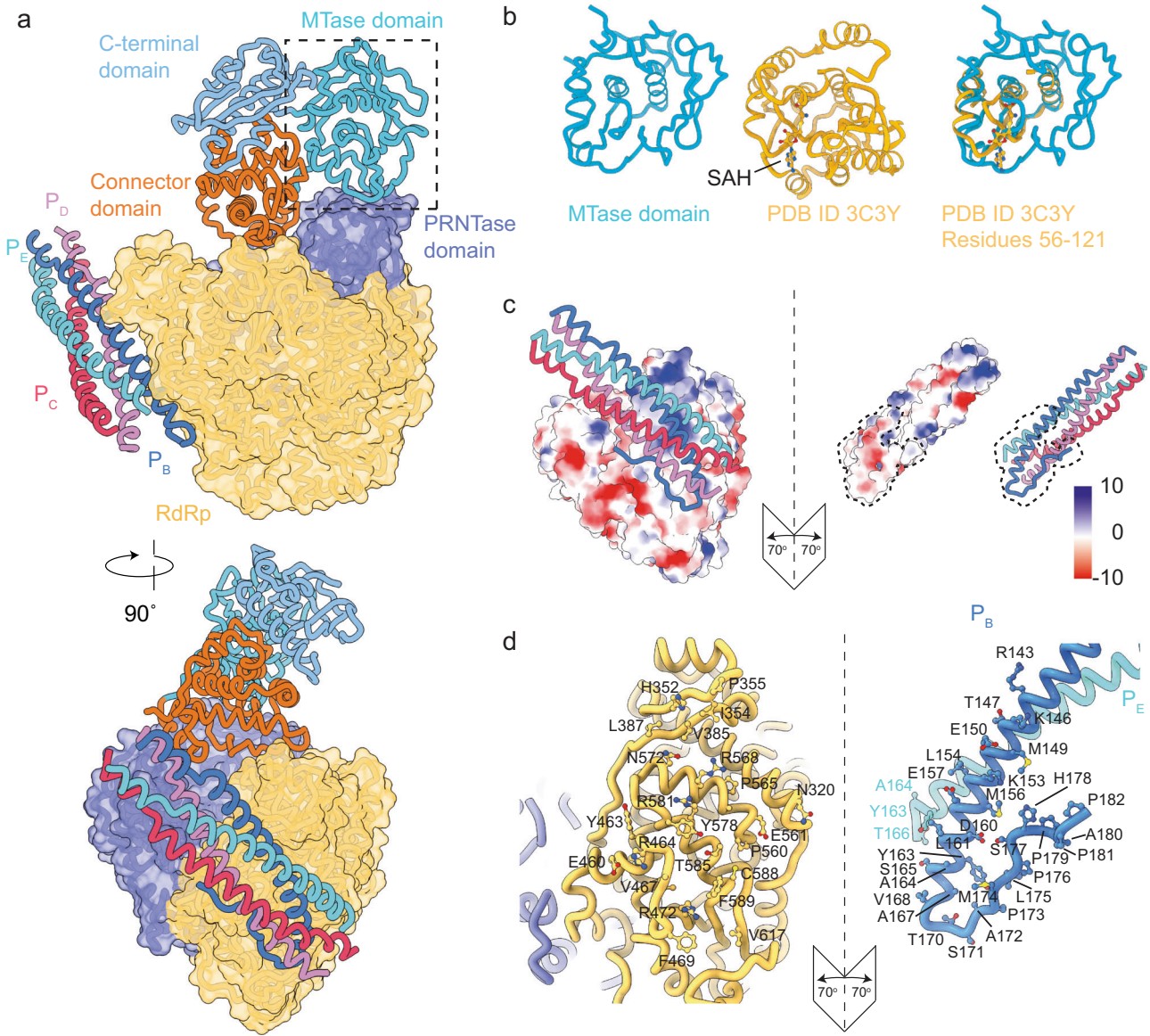

**Fig. 4 | Interactions of the BoDV-1 L-P complex. a** C-terminal domain annotation showing the position relative to the core. **b** structure alignment of the MTase domain from BoDV-1 with the MTase from Mesembryanthemum crystallinum (PDB ID 3C3Y). In the right panel, showing the overlay of the two models, only the structurally homologous region from 3C3Y is included. SAH molecule is included to identify the catalytic site. **c** electrostatic surface representation of the interface between the L-protein and P-protein. P-protein chains are coloured as in panel (**a**). **d** participating residues from the L-protein RdRp core and from chain P_B and P_E. Colouring as in panel (**a**). Chains P_C/P_D are hidden for clarity. The orientation of the proteins in panels (**c**) and (**d**) is the maintained between, panel (**d**) is zoomed to encompass the dashed line region in panel (**c**).

though it is significantly smaller. The filovirus structures bury a surface area of approximately 3200 Å$^2$ (Ebola virus, PDB ID 7YES), paramyxovirus structures between 2700 Å$^2$ (hPIV3, PDB ID 8KDC) and 3500 Å$^2$ (NDV, PDB ID 7YOU and NiV, PDB ID 9FUX), and pneumovirus structures between 4100 Å$^2$ (HMPV, PDB ID 6U5O) and 4500 Å$^2$ (RSV, PDB ID 6UEN). These large interaction interfaces are mostly mediated by ordered domains at the C-terminal X domain of the P-protein, forming additional interactions away from the tetrameric binding site. Unlike these viruses, BoDV-1 only contains 20 amino acids C-terminal to the ordered binding site, and does not contain an X domain. Rhabdoviruses, where the P-protein is small and largely disordered, bury between 1600 Å$^2$ (VSIV, PDB ID 6U1X) and 2000 Å$^2$ (RABV, PDB ID 6UEB) via a peptide that wraps around the L-protein's CTD[18,19].

## Conformation of the BoDV-1 L-protein

The core of nsNSV L-proteins, comprising the RdRp and PRNTase, remains globally conserved in their position in the observed

structures describing stages of replication and transcription. In contrast, the C-terminal region, comprising CD, MTase, and CTD, undergoes significant rearrangement during these processes. In our model, the C-terminal region of the L-protein is in an open arrangement and positioned away from the catalytic centres of the RdRp and PRNTase domains (Fig. 5a). This arrangement is much more open than those of previously determined nsNSV L-protein structures. Analysis of the nsNSV models revealed two groupings of positions of the C-terminal region. One group is formed by two structures from the *Rhabdoviridae* family (VSIV and RABV) and three from the *Paramyxoviridae* (MuV, hPIV3, and NDV), which show an approximately conserved domain position (Fig. 5a). In the *Rhabdoviridae* models, the CTD appears to be restrained by the presence of the P-protein, which is wrapped around these domains. In the *Paramyxoviridae* models, however, there appears to be no interaction between the C-terminal region and P-protein.

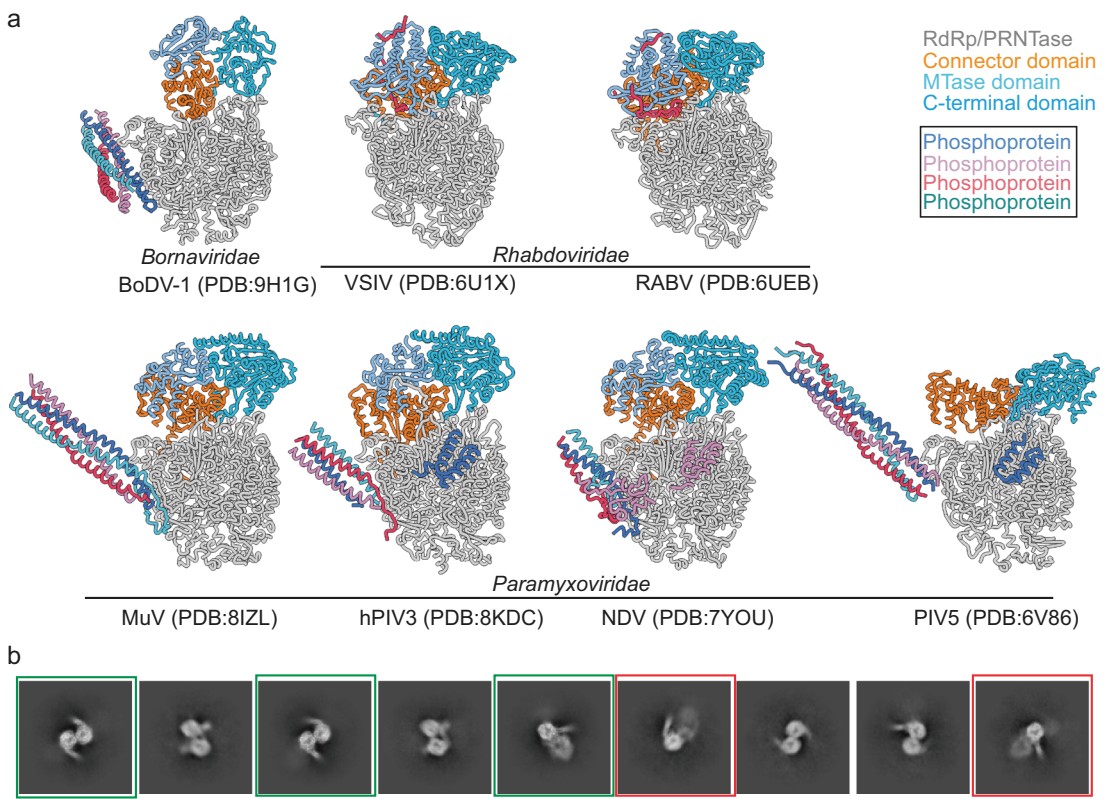

**Fig. 5 | Conformations of LP complexes and oligomerisation of the BoDV-1 LP.** **a** comparison of full-length L-protein structures from *Bornaviridae*, *Rhabdoviridae*, and *Paramyxoviridae*. The RdRp/PRNTase (grey), connector domain (orange), MTase domain (cyan), and C-terminal domain (blue) are shown for the L-protein. The four phosphoprotein molecules are shown in dark blue, pink, purple, and dark green. **b** 2D classes showing dimeric polymerase classes. Green boxes show well-ordered dimeric complexes and red boxes show complexes where only one polymerase appears well ordered.

A second group, comprising the PIV5 model, describes a drastically different arrangement where the CD has moved to the back of the RdRp and the MTase is brought proximal to the PRNTase active site. This conformation has been annotated as transcriptionally competent[11], and has presently only been observed for PIV5.

Functional annotation of these structures is dependant not only on the arrangement of the CTD position, but also on the arrangement of the PRNTase active site motifs, which control access to the MTase domain. In conformations where the MTase is not accessible by a nascent RNA product, these have been annotated as replicative or early transcriptive conformations.

## Oligomerisation of the BoDV-1 L-P complex

Detailed analysis of both cryoEM samples revealed minor populations of particles that contained additional map density of BoDV-1 L-proteins. In the RNA free-sample iterative rounds of classification from the consensus refinement revealed a set of particles, comprising approximately 2% of the total, which contained additional density between the CTD and RdRp (Supplementary Fig. 1, Supplementary Fig. 6a, b). Given the high purity of the sample and that we have been able to resolve the ordered regions from the single BoDV-1 LP complex, we tested whether single BoDV-1 L domains would fit in this density. Using the size and shape constraints of the density we concluded that the CTD was the best fit of the three domains in the C-terminal region (Supplementary Fig. 6c). Classification of these particles in a much larger box did not yield observations of full dimeric particles. This suggests a complex of BoDV-1 LP with an additional copy of the CTD can form.

Due to the low resolution of the map in this region, we were unable to uniquely orient the domain in the density or exclude that this is a contaminating protein from the expression cells present at low

concentration. Recent structural data on the hPIV3 L-protein showed an additional copy of the CD in the model[15] in an oligomeric assembly. However, this was located on a different surface of the RdRp domain, and the hPIV3 map was determined to a much higher resolution. While an interesting observation, we are unable to conclude the biological relevance of this observation.

Also present in this dataset were ~400 particles that were assigned to a tetrameric oligomer (Supplementary Fig. 1). As only one orientation of this assembly was observed, we were unable to reconstruct a map.

Careful analysis of the sample 2 dataset, by re-extraction of all particles in a larger box followed by multiple rounds of 2D classification, revealed a subset of particles which had formed a dimeric structure (Fig. 5b). In some classes only one L-protein showed features consistent with secondary structure while the second L-protein was blurry (Fig. 5b, red boxes), suggesting the dimer is not well ordered. In other cases, both L-proteins show clear secondary features suggesting a well-ordered dimer (Fig. 5b, green boxes). While some classes appear symmetric, others appear asymmetric, suggesting possibly multiple types of dimerisation. Repicking with dimeric particles as templates and several reconstruction/sorting strategies did not reveal additional views of these complexes, so unfortunately, reconstruction was not possible. We do not observe these oligomers in our mass photometry data, likely due to the low protein concentration and the oligomerisation being concentration-dependent.

Dimeric nsNSV L-proteins have been observed by low-resolution electron microscopy for VSIV L-protein[19] and more recently a partial dimeric structure has been observed at high-resolution by cryoEM for hPIV3[15]. For segmented RNA virus (sNSV) L-proteins, oligomeric complexes have been widely observed for the influenza virus[35–42] and for some bunyaviruses including Arenaviruses[43] and Hantaviruses[44]. These

complexes have been shown to be mediated by host proteins (Acidic Nuclear Protein 32 kDa for influenza virus)[35,41,42], viral proteins (Z-protein for Arenavirus)[43], or by direct L-protein - L-protein interactions (Influenza virus and Hantavirus)[36–39,44].

These oligomers are reported to have important roles in replication where they initiate/facilitate the capture of newly synthesised viral RNA during the formation of new RNP complexes[35,37,41,42]. Mutations in the Arenavirus L-protein dimer interface reduced function[43]. The role of higher-order oligomers observed in influenza virus polymerase[36] and hantaviruses[44] remains unclear.

Comparison of our two cryoEM datasets appears to reveal the formation of BoDV-1 dimers in response to the reaction conditions. In our sample 1 dataset we were able to identify a partial dimeric class interaction which we have putatively assigned as the CTD from a second molecule, we observe ~2% dimers and ~0.2% tetramers (Supplementary Fig. 1). In contrast, we observe ~6% of the particles as dimers in the sample 2 dataset (Supplementary Fig. 3b). From our present data, we are unable to identify the interface through which the dimer assembles. These data add BoDV-1 as a new example of nsNSV L-proteins that can oligomerise.

## Discussion

The complex of L- and P-proteins is required to perform transcription and replication of the viral genome in BoDV-1. Our data provide molecular information on the formation of the BoDV-1 replication complex and suggest some interesting differences to the existing paradigm for nuclear replicating nsNSV.

With the addition of a structure from the *Bornaviridae* family, there are now three families of the *Mononegavirales* (in addition to the *Filoviridae* and *Paramyxoviridae*) which demonstrate a tetrameric P-protein binding to an L-protein. While the angle of the P-protein oligomerisation domain and the L-protein is different between these families, the approximate interaction site with the L-protein is conserved (Fig. 5a). The function of the *P*-protein is likely conserved across nsNSV mediating interactions between the L-protein and RNA-free/bound *N*-protein[34]. Early studies on L-P-N interactions identified the extreme C-terminus of P-protein (residues 197–201) as important for binding N-protein[33]. This is consistent with our model, where these residues do not contact the L-protein and would be available to bind to a nucleocapsid assembly.

*Paramyxoviridae* are the most well-studied nsNSV system for understanding the molecular basis of L-protein movement along the nucleocapsid during replication and transcription[11–13,15,24,29]. In these models, the L-P interface is formed by multiple copies of P-protein forming unique contacts with L-protein. Commonly, this interaction interface in the P-protein is constituted by both the oligomerisation domain and multiple interactions from the ordered P-protein CTD. In contrast to these nsNSV models, in the BoDV-1 complex only a single copy of P-protein makes significant interaction with the RdRp. How this much smaller and singular interaction can maintain contact between the P-protein and L-protein during polymerisation will require further study.

The BoDV-1 models we present are distinct from other nsNSV L-proteins in the arrangement of their C-terminal region. However, since the PRNTase active site is not ordered, we cannot attribute this arrangement to a specific functional state. The *Bornaviridae* L-proteins do not contain the characteristic motif residues required for methylation of the transcription product, though our structural analysis reveals a fold which appears to be conserved with other MTase domains (Fig. 4b). As the L-protein of *Bornaviridae* likely does not perform methylation, the C-terminal regions of these proteins may not adopt similar conformational states due to differing functional requirements.

Given the pressure on genome size, it is intriguing that the virus has maintained these catalytically non-functional domains. Thogoto virus, a sNSV related to the influenza virus, has been recently shown to contain a cap-binding domain which had similarly lost its ability to bind the cap[45,46]. It may be that these domains are now performing important structural roles or contain interaction sites for protein partners and have therefore been maintained. How *Bornaviridae* can produce mRNA which are either methylated by nuclear host proteins or efficiently translated in the absence of methylation is currently unknown. Recently, proximity ligation data has suggested the L-protein interacts with host nuclear MTase proteins which may perform this function[47].

In summary, we have performed a detailed structural analysis of the replication complex of the BoDV-1 LP-complex and these findings strengthen our understanding of both this family and of the wider nsNSV grouping. These studies present a starting point for investigations into the nuclear stage of the *Bornaviridae* lifecycle and identify potential targets for antiviral therapeutic studies.

## Methods

### Protein cloning, purification, and mass photometry

The mammalian BoDV-1 *P*-protein (UniProt P0C799) with an N-terminal octaHis-tag and TEV-protease site and *L*-protein with a N-terminal twin-strep, octaHis-tag and TEV-protease site were cloned into separate cassettes in the pFL plasmid as part of the MultiBac system. Sf9 cells were grown in Sf-900 II serum-free media (Gibco). Three days after infection cells were harvested and resuspended in a wash buffer containing 50 mM HEPES, pH 7.5, 500 mM NaCl, 0.05% (w/v) n-Octyl beta-D-thioglucopyranoside, 2 mM dithiothreitol, 10% (v/v) glycerol, one protease inhibitor tablet (Sigma), 5 mg RNAse, and 2.4 mL of BioLock (IBA). Cells were disrupted by sonication and clarified with centrifugation. The supernatant was then applied to Strep-Tactin Superflow high-capacity (IBA) resin and incubated for 2 h. The resin was then washed with 50 column volumes of a buffer containing 50 mM HEPES, pH 7.5, 500 mM NaCl, 0.05% (w/v) n-Octyl beta-D-thioglucopyranoside, 2 mM dithiothreitol, and 5% (v/v) glycerol. Prior to elution the resin was washed with ten column volumes of 50 mM HEPES, pH 7.5, 500 mM NaCl, 2 mM dithiothreitol. Borna L-P complex was eluted with 100 mM Tris-HCl, pH8.0, 250 mM NaCl, 1 mM dithiothreitol, 5% (v/v) glycerol, and 50 mM Biotin. Protein sequence analysis was visualised using ESPript 3.0[48].

Mass photometry was performed on a TwoMP mass photometer (Refeyn LTD). Calibration was performed with three standards (thyroglobulin, ovalbumin and aldolase) each diltuted into the buffer used for BornaLP analysis. For sample analysis 1 μL BornaLP at a concentration of 1.2 mg/ml was diluted into a buffer containing 20 mM, HEPES, pH 7.5, 250 mM NaCl, 0.5 mM dithiothreitol, and 5% (v/v) glycerol. Data analysis was performed using Dis-coverMP software 2024 R2 (Refeyn Ltd).

### CryoEM sample preparation

For Sample 1, the protein was used immediately after elution to prepare cryoEM samples. Grids were prepared using a Vitrobot mark IV (FEI) at 100% humidity. Quantifoil Holey Carbon (R2/1, 200 mesh copper) grids were glow discharged, before a volume of 3.5 μL sample at 0.33 mg/ml was applied for 3.5 s before vitrification in liquid ethane.

Sample 2 was prepared by the addition of template RNA (5'-AAUGAUUGGGUUUGUUGUUAACGC-3') to a final concentration of 25 μM, a triphosphorylated primer (5'-pppGpCpGpUpUpA-3') to a final concentration of 20 μM, 1 mM S-Adenosyl methionine, and 1 mM each of ATP, GTP, CTP, UTP. The reaction was then incubated on ice for 1 h before being diluted to a final concentration of 0.4 mg/ml and grids prepared as before.

### CryoEM image collection and data processing

Cryo-EM data for both L-protein samples were collected at the Oxford Particle Imaging Centre (OPIC), on a 300 kV G3i Titan Krios microscope (Thermo Fisher Scientific) fitted with a SelectrisX energy filter

and Falcon IVi direct electron detector. Automated data collection was setup in EPU 3.6 and movies were recorded in EER format. Sample was collected with a total dose of ~ 50 e-/Å², a calibrated pixel size of 0.932 Å/pix and a 10 eV slit. Sample-specific data collection parameters are summarised in Supplementary Table 1.

## CryoEM data processing

The processing pipeline, particle distribution, and resolution ranges of the resulting reconstructions are summarised (Supplementary Fig. 1, Supplementary Fig. 3b). Both datasets were processed using cryosparc V4.0-4.5[49] using a common initial workflow. Movies in eer formation were fractionated into 60 frames prior to patch motion correction and patch CTF estimation using default settings. Movies of poor quality were manually removed. Initial particle picks were generated using the blob-picker followed by 2D classification to generate templates for the template-based picker.

For the sample 1 the initial picks were used to generate initial 3D volumes using ab initio model generation and heterogenous refinement. This initial model was highly preferentially oriented. This particle set was then used to train a model in Topaz[50] and generate a new particle set, which was then extracted and subject to 2D and 3D classification. This particle set generated a consensus refinement to a global resolution of 3.13 Å using non-uniform refinement and local refinement. This model was further classified using 3D classification to generate two particle sets that were refined using non-uniform refinement and local refinement. Tetrameric particles were observed through re-extraction in a large box followed by 2D classification.

For the sample 2, initial particle picks were of a sufficiently high quality to perform Topaz training and picking prior to ab initio model generation and heterogenous refinement. Consensus refinement of the best class generated a reconstruction to 2.83 Å. Subsequently, heterogeneous refinement revealed one class that contained an identical core to the RNA-free reconstruction. Heterogenous refinement of these particles revealed two related conformations one of which contained a full L-protein, which the other contained only the core and P-protein. In parallel, this large particle set was re-extracted in a large box and multiple rounds of 2D classification were performed to visualise particles that were present in dimeric assemblies.

## Structure determination and model refinement

An initial model was generated for the RNA-free BoDV-1 complex through in silico model prediction using Alphafold2 as implemented in CollabFold[26,51]. The model was then manually fitted in the map using UCSF ChimeraX V1.7-9[52], followed by iterative refinement in PHENIX V1.20[53] and Coot V 0.97[54]. The final model geometry and map-to-model correlation were validated using PHENIX Molprobity. Subsequent models were generated using this model as the template followed by iterative refinement as above in PHENIX and Coot. All maps and model statistics are summarised in Supplementary Table 1. All structural analysis and figures were generated using UCSF ChimeraX V1.7-1.9[52].

## In vitro RNA synthesis assay

For the BoDV LP complex in vitro assay, 3 µl reactions were set up containing reaction buffer [20 mM Hepes pH 7.5, 100 mM NaCl, 1 mM DTT, 5% (v/v) Glycerol, 1 U Rnasin (Promega), 5 mM MgCl], 2.5 µM of an 13 or 14 nt long RNA-template (pUUGUUAACGCAACp or pUU-GUUAACGCAACAp), 1 µM recombinant BoDV LP complex, 0.5 mM GTP, 0.5 mM CTP, 0.5 mM CTP and 0.5 µM ATP. The radioisotope tracer in these reactions was [α³²P] ATP (Revvity). The reactions were incubated at 30°C for 1 h, stopped with the addition of 3 µl formamide loading buffer and denatured at 95°C for 3 min. ³²P-5'end labelled 13 nucleotide-long RNA templates served as a marker. RNA products were separated on a 22% polyacrylamide urea gel for 2.5 h at 30 W and the level of [α³²P] AMP incorporation was visualised by phosphorimaging on an FLA-5000 scanner (Fuji).

## Reporting summary

Further information on research design is available in the Nature Portfolio Reporting Summary linked to this article.

## Data availability

The data supporting the findings of this study are available from the corresponding authors upon request. CryoEM maps and models generated in this study have been uploaded to the PDB and EMDB. The accession codes for the Sample 1-RNA Free L-protein are 9H1G /EMDB-51765 [https://www.ebi.ac.uk/emdb/entry/EMD-51765]). The accession codes for the Sample 2 core complex (9H1Q/EMDB-51770 [https://www.ebi.ac.uk/emdb/entry/EMD-51770]) and full complex (9H1Y/EMDB-51785[https://www.ebi.ac.uk/emdb/entry/EMD-51785]). The atomic coordinates published by other authors are available in the PDB for Ebola (7YES and 8JSN), hPIV3 (8KDC), NDV (7YOU and 9FUX), HMPV (6U5O), RSV (6UEN,), VSIV (6U1X), RABV (6UEB), and M. crystallinum MTase (3C3Y]). Source data are provided as a source data file.

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

## Acknowledgements

We thank Dr Esra Balikci-Akil and members of the Fodor and Grimes laboratories for helpful comments and discussions. This work was supported by Wellcome Investigator Award 200835/Z/16/Z (to J.M.G.) and by a BBSRC International Institutional Award (to J.R.K.). Computational aspects of this work were enabled by the Oxford Biomedical Research Computing (BMRC; DOI-2025) facility. For the purpose of open access, the author has applied a Creative Commons Attribution (CC-BY) licence to any Author Accepted Manuscript version arising from this submission. Correspondence and requests for materials should be addressed to J.R.K.

## Author contributions

L.C., F.G., A.D., J.M.G., and J.R.K. conceived the study, produced samples, collected data, analysed data, and wrote the paper.

## Competing interests

The authors declare no competing interests.
