## [Transparent Peer Review file · Nature Communications]

The structure of the mammalian bornavirus polymerase complex

Corresponding Author: Dr Jeremy Keown

Version 0:

Reviewer comments:

Reviewer #1

(Remarks to the Author)

In this manuscript, Carrique and colleagues present the cryo-EM structures of Borna disease virus 1 (BoDV-1) RNA polymerase (L) in complex with phosphoprotein (P). The authors successfully resolved both the full and partial structure of L. They observed four copies of P, but only one copy interacts with L, which is different from other resolved L/tetrameric P complexes. In addition to the native structure, the authors report another structure in which the complex, incubated in a buffer containing an RNA template, triphosphorylated primer, SAM, and NTPs, shows slight conformational changes. However, no RNA is observed in this structure.

This research adds to a series of recent studies describing the structures of non-segmented negative-sense RNA viruses. It describes some differences between the BoDV-1 L-P complex and earlier structures and provides the structural basis for the relevant antiviral development; however, it does not significantly enhance our understanding of how these RNA polymerases transcribe viral genes and replicate the viral genome. Additionally, this reviewer has several concerns that need to be addressed regarding this manuscript.

Major points:

1. There are no in vitro enzymatic assays or other functional data to confirm that the obtained BoDV-1 L-P complex is active.

2. Maps and models:

(1) In the PDB validation report, for 9H1G, the resolution reported by authors is 3.07 Å, but the resolution shown in author-provided FSC curve is 3.96 Å. Ramachandran outliers need to be fixed. The value of Sidechain outliers is poor. It's better to improve it.

The partial of the densities covering CD-MTase-CTD domains is poor, which makes it hard to dock the corresponding model or/and assign the side chains. The authors may get a better local map from local/focused refinement on this region.

(2) Similarly, in the PDB validation report, for 9H1Q, the resolution reported by authors is 2.95 Å, but the resolution shown in author-provided FSC curve is 3.82 Å. The value of Sidechain outliers is poor. It's better to improve it.

(3) In the PDB validation report, for 9H1Y, the resolution reported by authors is 3.07 Å, but the resolution shown in author-provided FSC curve is 4.05 Å. Ramachandran outliers need to be fixed. The value of Sidechain outliers is poor. It's better to improve it.

The local resolution of the densities covering CD-MTase-CTD domains is relatively low, which is much worse than that in EMD-51765. It's puzzling that the mean bfactor of all the atoms in 9H1Y is lower than that of 9H1G.

3. Fig. 2 and lines 160-172. The authors describe differences between the structures obtained with and without RNA. They speculate that these structural changes result from the addition of RNA, although no RNA is observed in the structure. However, the descriptions of these structural rearrangements are not entirely clear. What is the function of the residues that have changed? How might these rearrangements contribute to RNA replication? Furthermore, these changes could also result from other factors, such as the buffer composition or the inherent flexibility of the CD-MTase-CTD domains. Additional data are needed to support the claim that the conformational changes are due to RNA binding.

4. Fig. 4, S Fig. 5 and Lines 266-311. The authors describe the dimeric BoDV-1 L/P complexes. They state: "Given the high purity of the sample and considering that we could resolve all ordered regions from a single BoDV-1 L-P complex, we concluded that this extra density was likely coming from an additional copy of L-P. Considering the size of the density we conclude that this density is likely an additional copy of the CTD." However, it isn't easy to confidently assume that this additional density corresponds to the CTD of another copy since its local resolution is around 15 Å. The authors may consider re-processing the data by selecting dimeric particles and generating a map of the dimeric L-P complex. This approach could provide more convincing evidence regarding how L-P particles dimerize.

Minor points:

1. The term "replication complex" in the title is not entirely appropriate here, as replication is only one of the functions of the L/P complex.
2. Fig. 2 c. The rotation angle is not shown.
3. Supplementary Fig. 2. The range of different domains/subdomains should be marked.
4. The structural models are presented in the same way across all figures in the manuscript, making it difficult to visualize the secondary structure of the complexes clearly.

Reviewer #2

(Remarks to the Author)

Carrique et al. report the high-resolution cryo-EM structure of the borna disease virus 1 (BoDV-1) polymerase complex, which comprises the L protein in complex with its tetrameric phosphoprotein (P) co-factor. They observe that only one of the P protomers interacts with L. Visualizing samples in the process of RNA leads to conformational changes, whereby opening and closing of the template entry and exit channels are respectively observed. The authors also observed evidence for formation of oligomeric complexes. The pRNTase domain priming and intrusion loops are disordered in the structures, but the L C-terminal globular domains are observed. The authors made the interesting observation that structural features of the L MTase domain suggest that this portion of the protein lacks enzymatic activity. Overall, the structural work is of high quality, although certain functional analyses would be required to support and strengthen some of the observations.

Major.

1. The authors did not test the L protein they used for structural analysis for RNA synthesis activity. It is important to confirm that the protein used for structural analysis is functional and represents an active form.
2. One important observation is that the MTase domain of the L protein would not be structurally compatible with having enzymatic activity, although the authors did not test this directly. Experimental evidence to biochemical assays should be provided to support this notion.

Minor.

1. The paper could benefit from the addition of a phylogenetic tree (supplementary figure) of the Mononegavirales L protein rooted at the RdRp domain.
2. In Figure 3c, a scale should be included for the electrostatic potential that is shown.
3. Figure S1a – a scale bar should be added to the micrograph. Also, could the authors clarify the nature of the coloring scheme and the scale shown for the two volumes shown in the bottom right of the figure?
4. Figures S1 and S2. Please increase the size of the GSFSC and particle angular distribution plots. The fonts are too small to read at the current size.
5. Lines 148–150. "Within the Bornaviridae the BoDV-1 RdRp and PRNTase sequences appear well conserved, while the C-terminal regions show a much greater sequence diversity"
The authors should consider annotating figure S2 to include domains/domain boundaries. e.g., Are there certain subdomains that are less conserved?
6. Lines 162–163 and Figure 2c,d. "In the reaction structure, residues 727-735 from the RdRp have become disordered and residues 400-407 have retracted away from the active site." As presented, these structural changes are difficult to visualize. The authors should consider generating individual panels to show these particular changes for clarity.
7. Lines 170–172: "All attempts to capture RNA bound complexes were unsuccessful."
The authors should be more explicit about what they tried as part of their efforts.
8. Lines 308–309. "Comparison of our two cryoEM datasets appears to reveal the formation of BoDV-1 dimers in response to the reaction conditions". Could these authors clarify what they mean here, e.g., is there a quantitative assessment of the number of dimers in both datasets that can be provided?
9. Lines 346–348: "Thogoto virus, a sNSV related to the influenza virus, has been recently shown to contain a cap-binding domain which had similarly lost its ability to bind the cap"
It seems like the following reference should be included here – PMID: 24454773.
10. The authors comment on possible oligomerization states (dimers and tetramers) observed in their cryoEM datasets but did not observe species of these sizes in their mass photometry experiment. Could the authors comment on this? Is it possible that the oligomerization is concentration-dependent? This should be addressed in the discussion.

Reviewer #3

(Remarks to the Author)

The manuscript provides a comprehensive structural analysis of the Borna disease virus 1 (BoDV-1) replication complex, a critical component of the virus's nuclear-replicating lifecycle. Using cryo-electron microscopy, the authors resolved high-quality structures of the viral L-protein (RNA-dependent RNA polymerase) in complex with the P-protein (phosphoprotein) at resolutions up to 2.8 Å. The study highlights distinct features of BoDV-1, including its relatively smaller L-protein and P-protein compared to other non-segmented negative-sense RNA viruses (nsNSVs), while also demonstrating significant structural similarities consistent with other nsNSVs.

While the manuscript is well-written and the structural data are robust, it is limited by its lack of functional analysis. Key assays, such as polymerase activity tests, RNA 5' end capping assays, or cell-based studies to dissect the functional implications of unique structural features, are absent. Without these experimental validations, the conclusions drawn remain largely descriptive and lack sufficient depth to fully elucidate the biological relevance of the findings.

Specific comments:

1. The authors should consider performing assays to detect RNA polymerase activity and RNA capping products. Established methods from published studies on viral RdRps and capping enzymes are available and could be adapted for this purpose.
2. The RNA substrate used in the reconstitution study is only described textually. It would enhance clarity and understanding to include a figure illustrating the RNA sequence and its predicted base-pairing pattern, such as in Figure 2.
3. The authors could explore modeling the RNA substrate into the observed "open" conformation of the structure. This could help identify potential conformational changes required for RNA accommodation and provide insights into the dynamics of the replication process.
4. The multiple sequence alignment (MSA) could be expanded to include replicase sequences from other nsNSVs. This broader comparison would help place the structural features of BoDV-1 in a more comprehensive evolutionary and functional context.
5. It remains unclear whether the 5' end of the BoDV-1 genome is capped, and if so, whether it is a cap 0 or cap 1 structure. Similarly, there is limited information about the capping status of the viral mRNAs. The authors should discuss whether there is any evidence suggesting that BoDV-1 might utilize host mRNA capping machinery, as this could have significant implications for the viral replication strategy.

Reviewer #4

(Remarks to the Author)

Version 1:

Reviewer comments:

Reviewer #1

(Remarks to the Author)

The authors provide additional functional validation of the L–P complex in the revised submission, addressing some prior concerns. However, the revision also introduces new descriptions that raise further questions and warrant clearer explanation.

Unresolved concern:

1. The manuscript's descriptions of secondary structure elements are inconsistent and imprecise. For instance, Line 135 mentions a β -strand; Lines 214–215 cite two antiparallel α -helices and a large β -sheet; Lines 220–221 reference two central α -helices, two β -strands, and an additional α -helix; and Line 260 refers to a "domain helix." However, the current licorice-style visualizations do not distinguish these features from flexible loops, making them difficult to interpret. Clearer annotation or improved structural visualization is needed to support the manuscript's claims.

Additional concerns:

1. The authors use 13-nt and 14-nt RNA templates in their in vitro polymerase assays and suggest, based on their data and prior studies, that initiation occurs at a cytidine residue located at the third or fourth nucleotide from the 5' end (supplementary figure 2). This conclusion overstates the evidence. While internal initiation is not without precedent, the claim that the first 3–4 nucleotides are non-functional as template bases is speculative and insufficiently supported by experimental data. Further validation is needed to justify this interpretation.
2. In the text (line 141), the authors claim that the 3–4 bases at the 3' end of the template RNA are essential for polymerase activity. However, these bases were omitted in the RNA template used for the proposed reaction complex structure. Moreover, the in vitro assays do not confirm that this RNA in the structure study is catalytically competent. Referring to the structure as a "reaction complex" is therefore misleading and unsupported by the current evidence, especially if the omitted bases are indeed critical for function.

Reviewer #2

(Remarks to the Author)

Overall, the authors have addressed my concerns. The inclusion of the RNA-dependent RNA polymerase activity assay results strengthens the manuscript and increases the biological relevance of the structures the authors report.

I have a minor comment related to another reviewer's comment, but I think that it is important to comment on it nonetheless. The authors write in their rebuttal that "A recent manuscript (PMID: 38605025) describes an extra copy of the CTD bound to an nsNSV L protein, but no full dimeric complex." I believe the authors here refer to Xie et al. Nat Comm 2024, which reports the cryo-EM structure of hPIV3 L-P in complex with the connector domain of a second L copy, but not in complex with an extra copy of the CTD, as noted by the authors?

This would suggest a difference – Bornavirus L potentially interacts with the C-terminal domain of a second L protomer while hPIV3 L interacts with the connector domain of a second L protomer. How certain are the authors that the extra domain/density they observed in the Bornavirus L-P map is a second C-terminal domain as opposed to a second connector domain? Would the connector domain fit or not fit in terms of its volume/shape? The authors should consider clarifying in the manuscript the level of certainty with which they claim the extra density they observe is accounted for by an additional copy of the C-terminal domain.

Reviewer #3

(Remarks to the Author)

The revised manuscript has addressed my concerns.

Suggestion: Move the polymerase activity data (Suppl Fig2) to main text, given its importance.

Reviewer #4

(Remarks to the Author)

REVIEWER COMMENTS

Reviewer #1 (Remarks to the Author):

In this manuscript, Carrique and colleagues present the cryo-EM structures of Borna disease virus 1 (BoDV-1) RNA polymerase (L) in complex with phosphoprotein (P). The authors successfully resolved both the full and partial structure of L. They observed four copies of P, but only one copy interacts with L, which is different from other resolved L/tetrameric P complexes. In addition to the native structure, the authors report another structure in which the complex, incubated in a buffer containing an RNA template, triphosphorylated primer, SAM, and NTPs, shows slight conformational changes. However, no RNA is observed in this structure.

This research adds to a series of recent studies describing the structures of non-segmented negative-sense RNA viruses. It describes some differences between the BoDV-1 L-P complex and earlier structures and provides the structural basis for the relevant antiviral development; however, it does not significantly enhance our understanding of how these RNA polymerases transcribe viral genes and replicate the viral genome. Additionally, this reviewer has several concerns that need to be addressed regarding this manuscript.

Major points:

1. There are no in vitro enzymatic assays or other functional data to confirm that the obtained BoDV-1 L-P complex is active.

During revision of our manuscript we have sought to expand our study to incorporate additional functional characterisation of our polymerase complex. Our key piece of additional data is a demonstration that the BornaLP complex is active on both a 13mer and 14mer RNA templates. These templates contain a three or four base extension at the 3' termini and are 13 bases shorter at the 5' termini, compared with our previous 23mer RNA template. The design of these alternate RNA are based on two earlier reports PMID: 15858040 and PMID: 21482759 which show a 5'-...AACCA-3' or 5'-...AAC-3' are important for extension.

Our 14mer template (5' pUUGUUAACG**CAAC**Ap 3') and the 13mer template (5' pUUGUUAACG**CAAC**p 3') both contain this 3' extension. Extension on either template shows identical products are formed on each, suggesting initiation is not occurring at the first nucleotide. These assays use 32pATP. Our reactions show a very strong product at 6nt (corresponding to the first incorporated A) with a decreasing abundance of products out to a final length of 10nt. From these data we conclude that the polymerase appears to be initiating opposite the 4/5th nt, a C shown in bold in the above templates. Studies on other nsNSV L-proteins for example ebola virus (PMID: 37699521) have shown similar extension patterns in in vitro assays with shorter early termination products predominating.

We have now included this data in supplementary figure 2, with a discussion of this data at lines 135-143, and methods section in lines 465-475.

Additionally, we have now included a detailed discussion (lines 243-254) of the L-P interface. Briefly minireplicon luciferase reporter assays and immunoprecipitation

experiments (PMID: 15509569 and PMID: 9535888) perfectly recapitulate the details of the structural analysis of this interface. Importantly residues 172-183, which we observe to be ordered in one P-protein model to form ~50% of the interface with the L-protein, are vital for the L-protein activity. We believe the incorporation of this discussion further validates the biological importance of this interface, in particular the importance of these additional ordered residues C-terminal to the oligomerisation helix.

2. Maps and models:

(1) In the PDB validation report, for 9H1G, the resolution reported by authors is 3.07Å, but the resolution shown in author-provided FSC curve is 3.96 Å. Ramachandran outliers need to be fixed. The value of Sidechain outliers is poor. It's better to improve it. The partial of the densities covering CD-MTase-CTD domains is poor, which makes it hard to dock the corresponding model or/and assign the side chains. The authors may get a better local map from local/focused refinement on this region.

This mismatch in the FSC in the PDB validation report is caused by the unmasked FSC being uploaded, we have updated the PDB with the masked FSC and the values now match. We have investigated and fixed rotamer/Ramachandran outliers with the new coordinates upload to the PDB. New model statistics have been included in supplementary table 1.

While we agree that the C-terminal regions of the L-protein are more poorly ordered than the RdRp/PRNTase domains, however we are confident in the position of the domains and backbone trace. The fitting of these domains is based on Alphafold predictions for which the domain structures/sidechains are high confidence. Our models were prepared by splitting the predicted model, fitting each domain, and manually relinking the domains. We agree that the accuracy of the many sidechains in the C-terminal region is low, for this reason we do not discuss interactions of sidechains in this region so to not over interpret our data.

To improve the density of the C-terminal region we have tried several approaches implemented in the leading cryoEM software processing packages including Relion5 (DynaMight and Blush) and Cryosparc (local refinement and flexible refinement). Subtraction of the RdRp/PRNTase region followed by focused refinement of the C-terminal region did not significantly improve the map.

(2) Similarly, in the PDB validation report, for 9H1Q, the resolution reported by authors is 2.95Å, but the resolution shown in author-provided FSC curve is 3.82 Å. The value of Sidechain outliers is poor. It's better to improve it.

This mismatch in the FSC in the PDB validation report is caused by the unmasked FSC being uploaded, we have updated the PDB with the masked FSC and the values now match. We have improved our model and fixed the rotamers as suggested. Updated model statistics have been included in Supplementary table 1.

(3) In the PDB validation report, for 9H1Y, the resolution reported by authors is 3.07Å, but the resolution shown in author-provided FSC curve is 4.05 Å. Ramachandran outliers need to be fixed. The value of Sidechain outliers is poor. It's better to improve it. The local resolution of the densities covering CD-MTase-CTD domains is relatively

low, which is much worse than that in EMD-51765. It's puzzling that the mean bfactor of all the atoms in 9H1Y is lower than that of 9H1G.

This mismatch in the FSC in the PDB validation report is caused by the unmasked FSC being uploaded, we have updated the PDB with the masked FSC and the values now match. We have investigated and fixed rotamer/Ramachandran outliers with the new coordinates upload to the PDB. New model statistics have been included in supplementary table 1.

We are unsure why this discrepancy in the mean-bfactor between the two models was present, upon reprocessing our models the mean values (193 vs 188) are now much closer as expected.

3. Fig. 2 and lines 160-172. The authors describe differences between the structures obtained with and without RNA. They speculate that these structural changes result from the addition of RNA, although no RNA is observed in the structure. However, the descriptions of these structural rearrangements are not entirely clear. What is the function of the residues that have changed? How might these rearrangements contribute to RNA replication? Furthermore, these changes could also result from other factors, such as the buffer composition or the inherent flexibility of the CD-MTase-CTD domains. Additional data are needed to support the claim that the conformational changes are due to RNA binding.

In response to this reviewers comment and point 6 from reviewer 2 we have now included an additional panels in figure 2 to help illustrate the movements we observe between the two samples. We think it is important that we do not conclude these changes are RNA induced, as the reviewer states. What we do conclude is that the conformational changes are caused by the reaction conditions, and these changing conditions correlate with an opening/closing of the template entry/exit channel of the polymerase core. As described in the original manuscript "Collectively, these movements in the PRNTase domain and RdRp have caused an expansion of the template entry channel such that a modelled template from the recent Ebola virus L-protein structural model could be accommodated without clashes and a tightening of the template exit channel.". Without being able to observe RNA entering/exiting the polymerase core we are unable to conclude more at this stage.

A similar phenomenon, where RNA and reaction conditions triggered a change in the nsNSV RdRp but RNA was not observed in the complex, was seen for the Ebola virus L-protein (PMID: 36171293).

We do not believe the difference in these residues is related to the flexibility of the C-terminal region. In the reaction dataset we observe and reconstruct models with the flexible region ordered (PDB 9H1Y) and absent (9H1Q), and the template entry/exit channels are in the same position in both structures.

We do not believe the difference is caused by just the buffer conditions, instead an active process has occurred in a subset of particles. In the reaction dataset we observe particles both in the apo conformation (90k particles, ~20% of picked particles) and reaction conformation (380k particles, ~80% of picked particles). As both conformations are present under the same buffer conditions, but only a subset of

particles have changed conformation, we conclude this is not related to the buffer composition.

4. Fig. 4, S Fig. 5 and Lines 266-311. The authors describe the dimeric BoDV-1 L/P complexes. They state: "Given the high purity of the sample and considering that we could resolve all ordered regions from a single BoDV-1 L-P complex, we concluded that this extra density was likely coming from an additional copy of L-P. Considering the size of the density we conclude that this density is likely an additional copy of the CTD." However, it isn't easy to confidently assume that this additional density corresponds to the CTD of another copy since its local resolution is around 15 Å. The authors may consider re-processing the data by selecting dimeric particles and generating a map of the dimeric L-P complex. This approach could provide more convincing evidence regarding how L-P particles dimerize.

We perhaps believe that the reviewer is combining the two different types of oligomerisation we observe. In Fig 4b, from our reaction dataset, we observe several well-ordered dimeric particle arrangements. These classes appear to represent different types of dimers, though attempts to reconstruct 3D volumes do not yield interpretable maps.

In our RNA free dataset, we can reconstruct a monomeric LP complex, and through 3D classification observe a map with additional density. This map is reconstructed from ~3,000 particles. We have performed extensive classification of this particle subset (and repicking with Topaz) in a large box to identify the full dimeric species, however we have been unsuccessful in this. We are not sure if these particles are low abundance or if this extra CTD is a partial L-protein. A recent manuscript (PMID: 38605025) describes an extra copy of the CTD bound to an nsNSV L protein, but no full dimeric complex. What we believe we can conclude is that BornalP forms dimers, and that this type of assembly is consistent with observation for many other segmented and non-segmented negative sense RNA viruses.

Minor points:

1. The term "replication complex" in the title is not entirely appropriate here, as replication is only one of the functions of the L/P complex.

We have replaced replication with polymerase. The title now reads "The structure of the mammalian bornavirus polymerase complex".

2. Fig. 2 c. The rotation angle is not shown.

We have updated this figure to include the angle and removed the eye/arrow and updated the figure legend to reflect this.

3. Supplementary Fig. 2. The range of different domains/subdomains should be marked.

We have added this annotation with colouring to match the manuscript.

4. The structural models are presented in the same way across all figures in the manuscript, making it difficult to visualize the secondary structure of the complexes clearly.

Throughout the manuscript we present the BornaLP using the licorice rendering in ChimeraX as we believe this provides an accurate yet visually clear description of our model. We do not believe that showing the BornaLP as the ribbon representation, to more definitively show the secondary structure elements, will increase the readability or perception of our manuscript.

Reviewer #2 (Remarks to the Author):

Carrique et al. report the high-resolution cryo-EM structure of the borna disease virus 1 (BoDV-1) polymerase complex, which comprises the L protein in complex its tetrameric phosphoprotein (P) co-factor. They observe that only one of the P protomers interacts with L. Visualizing samples in the process of RNA leads to conformational changes, whereby opening and closing of the template entry and exit channels are respectively observed. The authors also observed evidence for formation of oligomeric complexes. The pRNTase domain priming and intrusion loops are disordered in the structures, but the L C-terminal globular domains are observed. The authors made the interesting observation that structural features of the L MTase domain suggest that this portion of the protein lacks enzymatic activity. Overall, the structural work is of high quality, although certain functional analyses would be required to support and strengthen some of the observations.

Major.

1. The authors did not test the L protein they used for structural analysis for RNA synthesis activity. It is important to confirm that the protein used for structural analysis is functional and represents an active form.

Please see point 1 from reviewer 1 where we demonstrate the BornaLP complex we purify is active and include a detailed discussion of the L-P interface importance using literature results.

2. One important observation is that the MTase domain of the L protein would not be structurally compatible with having enzymatic activity, although the authors did not test this directly. Experimental evidence to biochemical assays should be provided to support this notion.

We have performed a MTase-Glo™ Methyltransferase Assay (Promega). In this assay we provided the BornaLP with a short pppGCGUUA primer, SAM, GTP which should enable the BornaLP PRNTase domain to first cap the primer and, if active, the MTase domain would the methylate this cap. Vaccinia capping system (NEB) was used as a well documented control MTase. We saw no significant activity. As this was a negative result, and at initial submission we had been unable to demonstrate even RdRp activity, we chose to exclude this result.

Minor.

1. The paper could benefit from the addition of a phylogenetic tree (supplementary figure) of the Mononegavirales L protein rooted at the RdRp domain.

As suggested, we have performed a structure based analysis (Foldmason, doi: <https://doi.org/10.1101/2024.08.01.606130>) to analyse experimentally determined non-segmented RNA virus L-proteins. This analysis clustered the L-proteins into their viral families, with Bornavirus closest to the root of the tree.

We have included this as Supplementary figure 6 and introduce it in the text as “A structure based phylogenetic analysis was performed using the Foldmason server³³ using representatives of the experimentally determined L-protein structural models (Supplementary Fig. 6). This phylogenetic tree clustered the seven *Paramyxoviridae* and two *Rhabdoviridae* structures together, with the *Bornaviridae* closest to the root of the tree.”.

2. In Figure 3c, a scale should be included for the electrostatic potential that is shown.

We have added the scale bar and annotated.

3. Figure S1a – a scale bar should be added to the micrograph. Also, could the authors clarify the nature of the coloring scheme and the scale shown for the two volumes shown in the bottom right of the figure?

We have added the scale bar and added “Map resolution (Å)” to clarify the panel in the bottom right.

4. Figures S1 and S2. Please increase the size of the GSFSC and particle angular distribution plots. The fonts are too small to read at the current size.

We have updated both cryoEM processing workflows, adding annotations and increasing figure sizes as requested.

5. Lines 148–150. “Within the Bornaviridae the BoDV-1 RdRp and PRNTase sequences appear well conserved, while the C-terminal regions show a much greater sequence diversity”
The authors should consider annotating figure S2 to include domains/domain boundaries. e.g., Are there certain subdomains that are less conserved?

As for minor point 3 from reviewer 1 we have added domain boundaries to this figure. In response to point 4 by reviewer 3 and to provide a more informative figure allowing comparison between other pathogenic nsNSV L-proteins we have now updated this figure to include a comparison to the Nipah, Ebola, and Rabies viruses.

6. Lines 162–163 and Figure 2c,d. “In the reaction structure, residues 727-735 from the RdRp have become disordered and residues 400-407 have retracted away from the active site.” As presented, these structural changes are difficult to visualize. The authors should consider generating individual panels to show these particular changes for clarity.

We have now added two panels to this figure as figure 2e and a zoom. We have rotated the presented model and removed some complicating regions of the model to more clearly show how these regions of the RdRp have rearranged.

7. Lines 170–172: “All attempts to capture RNA bound complexes were unsuccessful.” The authors should be more explicit about what they tried as part of their efforts.

We now include a brief description of our trialled conditions as follows “Our attempts to capture RNA bounds complexes, using the 14mer or 23mer RNA at 10X molar concentrations to BoDV-1 L-protein, in the presence of the 6mer primer, or with NTP at low sodium chloride concentrations was unsuccessful.”

8. Lines 308–309. “Comparison of our two cryoEM datasets appears to reveal the formation of BoDV-1 dimers in response to the reaction conditions”. Could these authors clarify what they mean here, e.g., is there a quantitative assessment of the number of dimers in both datasets that can be provided?

In the RNA free sample we observe ~2% dimers and ~0.2% tetramers. In the reaction complex sample we observe ~6% dimeric particles. What we are unable to conclude from our data is whether these are the same dimer in each dataset due to severe preferential orientation. From the reaction dataset it appears there are at least two different dimeric species present. We have added the below text in our manuscript “In our RNA free dataset we were able to identify a partial dimeric class interaction which we have putatively assigned as the CTD from a second molecule, we observe ~2% dimers and ~0.2% tetramers (Supplementary Fig. 1). In contrast we observe ~6% of the particles as dimers in the reaction complex dataset (Supplementary Fig. 3).”

9. Lines 346–348: “Thogoto virus, a sNSV related to the influenza virus, has been recently shown to contain a cap-binding domain which had similarly lost its ability to bind the cap”
It seems like the following reference should be included here – PMID: 24454773.

We have included this reference.

10. The authors comment on possible oligomerization states (dimers and tetramers) observed in their cryoEM datasets but did not observe species of these sizes in their mass photometry experiment. Could the authors comment on this? Is it possible that the oligomerization is concentration-dependent? This should be addressed in the discussion.

We believe this is being caused by the relatively low abundance of these species (0.2-6% of picked particles) and the dilute conditions under which the mass photometry is being performed. Mass photometry is conducted at approximately 1/10th of the concentration at which grids are prepared. We have added the below text to the manuscript

“We do not observe these oligomers in our mass photometry data, likely due to the low protein concentration and the oligomerisation being concentration dependant.”

Reviewer #3 (Remarks to the Author):

The manuscript provides a comprehensive structural analysis of the Borna disease virus 1 (BoDV-1) replication complex, a critical component of the virus's nuclear-replicating lifecycle. Using cryo-electron microscopy, the authors resolved high-quality structures of the viral L-protein (RNA-dependent RNA polymerase) in complex with the P-protein (phosphoprotein) at resolutions up to 2.8 Å. The study highlights distinct features of BoDV-1, including its relatively smaller L-protein and P-protein compared to other non-segmented negative-sense RNA viruses (nsNSVs), while also demonstrating significant structural similarities consistent with other nsNSVs.

While the manuscript is well-written and the structural data are robust, it is limited by its lack of functional analysis. Key assays, such as polymerase activity tests, RNA 5' end capping assays, or cell-based studies to dissect the functional implications of unique structural features, are absent. Without these experimental validations, the conclusions drawn remain largely descriptive and lack sufficient depth to fully elucidate the biological relevance of the findings.

Specific comments:

1. The authors should consider performing assays to detect RNA polymerase activity and RNA capping products. Established methods from published studies on viral RdRps and capping enzymes are available and could be adapted for this purpose.

Please see our response to point 1 from reviewer 1. Briefly, we have now established a templated extension assay which demonstrates the BornaLP sample we purify is active.

2. The RNA substrate used in the reconstitution study is only described textually. It would enhance clarity and understanding to include a figure illustrating the RNA sequence and its predicted base-pairing pattern, such as in Figure 2.

We have now included a scheme as panel Supplementary figure 3a above the cryoEM processing scheme for the reaction dataset. We believe this clearly illustrates our design for the reaction and avoids presenting it in the main body of the manuscript which may confuse the reader as we did not capture this complex. We have updated the figure legend and added appropriate references to this new figure throughout the manuscript.

In the revised manuscript we now include a *in vitro* assay (supplementary figure 2) and similarly display the RNA and reaction scheme above this figure for clarity.

3. The authors could explore modeling the RNA substrate into the observed "open" conformation of the structure. This could help identify potential conformational changes required for RNA accommodation and provide insights into the dynamics of the replication process.

This is an interesting suggestion and one which we have explored, especially in light of our new result showing how bases at the 3' end of the template RNA appear to be important for activity. These assays (presented in supplementary figure 2) show the template ending in ...AACGC is not extended, while AACGCAAC or AACGCAACA can be. On these longer templates the polymerase will still initiate from the **C** residue shown above. We suspect the AAC or AACA is important to correctly position the **C** in the active site. AF3 modelling with various sequences of template RNA paired short product RNA or NTP was not able to offer an explanation or possible hypothesis. While this manuscript was under review a structure of the Nipah virus LP- complex bound to RNA which mimics an elongation complex was published (PMID: 40050611). Alignment and comparison of this RdRp and RNA to aid in possible positioning of the RNA template did also not inform the on importance of these residues. Our future work will look to explore the early stages of transcription/replication initiation.

We have added a reference to this Nipah-RNA complex paper to our manuscript.

4. The multiple sequence alignment (MSA) could be expanded to include replicase sequences from other nsNSVs. This broader comparison would help place the structural features of BoDV-1 in a more comprehensive evolutionary and functional context.

We have now updated the alignment in supplementary figure 2 to include a comparison to several nsNSV L-proteins from other important viral species.

5. It remains unclear whether the 5' end of the BoDV-1 genome is capped, and if so, whether it is a cap 0 or cap 1 structure. Similarly, there is limited information about the capping status of the viral mRNAs. The authors should discuss whether there is any evidence suggesting that BoDV-1 might utilize host mRNA capping machinery, as this could have significant implications for the viral replication strategy.

From a structural perspective we observe the residues in the PRNTase domain required to perform capping. While these are not ordered in our models and we have not been able to demonstrate in vitro the capping ability of our LP sample we believe it likely the Bornavirus will produce a capped transcription product.

In contrast, the putative MTase appears to lack the necessary catalytic residues so we do not believe this will be active. As for the point by reviewer 2, we believe we have demonstrated the MTase is not active but at initial submission choose not to include this data as we had been unable to demonstrate RdRp activity, initially suggesting a possible defect with our sample.

We have investigated the literature to identify studies describing interactions between bornavirus and the host capping machinery. There are several studies describing interactions between the virus (specifically the L-protein) and it being tethered to the chromatin of infected cells. The initial submission included the statement "Recently, proximity ligation data has suggested the L-protein interacts with host nuclear MTase proteins which may perform this function⁴⁷.". Ref 47 (PMID: 34324219) describes the interaction with TRMT112, a host protein which is activator several host MTase, though the authors were unable to identify a specific MTase activity.

We are not aware of further studies which would be appropriate to include in our discussion.

Reviewer #1 (Remarks to the Author):

The authors provide additional functional validation of the L–P complex in the revised submission, addressing some prior concerns. However, the revision also introduces new descriptions that raise further questions and warrant clearer explanation.

Unresolved concern:

1. The manuscript's descriptions of secondary structure elements are inconsistent and imprecise. For instance, Line 135 mentions a β -strand; Lines 214–215 cite two antiparallel α -helices and a large β -sheet; Lines 220–221 reference two central α -helices, two β -strands, and an additional α -helix; and Line 260 refers to a “domain helix.” However, the current licorice-style visualizations do not distinguish these features from flexible loops, making them difficult to interpret. Clearer annotation or improved structural visualization is needed to support the manuscript's claims.

We have updated figure panels 2b to show the secondary structure elements in the classical ribbon annotation. In figure panel 4b, a zoom showing the putative MTase domain, we have similarly updated the figure to show ribbon annotation. We believe the oligomerisation domain helix referred to on line 260 is very clear in the current representation in figure panel 4d and does not require updating. Figure 4 was previously figure 3.

Additional concerns:

1. The authors use 13-nt and 14-nt RNA templates in their in vitro polymerase assays and suggest, based on their data and prior studies, that initiation occurs at a cytidine residue located at the third or fourth nucleotide from the 5' end (supplementary figure 2). This conclusion overstates the evidence. While internal initiation is not without precedent, the claim that the first 3–4 nucleotides are non-functional as template bases is speculative and insufficiently supported by experimental data. Further validation is needed to justify this interpretation.

We agree with the reviewers comments here that we are unable to conclusively determine the mechanism/mode of internal initiation from the current experiments. We have chosen to rewrite the section describing the assay, highlighting that we observe internal initiation and this has previously been observed as important for replication. We refrain from suggesting the mechanism for this or stating the bases are not functional as templates.

“To assess the enzymatic activity of the purified BoDV-1 L-P complex, we performed in vitro RNA synthesis assays using 13- and 14-nucleotide (nt) templates derived from the leader sequence (Fig. 3). The templates produced similar products with a strong signal at 6 nt and the weakest at 10 nt. From the pattern of radionucleotide incorporation we propose that these reactions have initiated internally on the template at positions 4 and 5 of the 13- and 14-nt templates, respectively. Previous work has shown that BoDV vRNA and cRNA promoters contain essential nontemplated extensions at the 3' termini, and that these are required for initiating RNA synthesis²⁸. Our results are consistent with these previous observations, supporting that initiation on these templates occurs after the 5'...AACAA-3' sequence at the 3'

termini. However, these extensions were not copied in our assays, suggesting their role may be important in positioning the template. The underlying mechanism by which the L-P complex restricts RNA synthesis to internal initiation or truncated products remains unclear.”

2. In the text (line 141), the authors claim that the 3–4 bases at the 3’ end of the template RNA are essential for polymerase activity. However, these bases were omitted in the RNA template used for the proposed reaction complex structure. Moreover, the in vitro assays do not confirm that this RNA in the structure study is catalytically competent. Referring to the structure as a “reaction complex” is therefore misleading and unsupported by the current evidence, especially if the omitted bases are indeed critical for function.

To remove any confusion, we now refer to the Apo sample as sample 1 and the reaction complex as sample 2. How these complexes were prepared is described in detail when they are first introduced.

Reviewer #2 (Remarks to the Author):

Overall, the authors have addressed my concerns. The inclusion of the RNA-dependent RNA polymerase activity assay results strengthens the manuscript and increases the biological relevance of the structures the authors report.

We thank the reviewer for their positive assessment of our work.

I have a minor comment related to another reviewer’s comment, but I think that it is important to comment on it nonetheless. The authors write in their rebuttal that “A recent manuscript (PMID: 38605025) describes an extra copy of the CTD bound to an nsNSV L protein, but no full dimeric complex.” I believe the authors here refer to Xie et al. Nat Comm 2024, which reports the cryo-EM structure of hPIV3 L-P in complex with the connector domain of a second L copy, but not in complex with an extra copy of the CTD, as noted by the authors?

Yes, the manuscript we are referring to is Xie et al. That is our mistake, and we have updated in our manuscript.

This would suggest a difference – Bornavirus L potentially interacts with the C-terminal domain of a second L protomer while hPIV3 L interacts with the connector domain of a second L protomer. How certain are the authors that the extra domain/density they observed in the Bornavirus L-P map is a second C-terminal domain as opposed to a second connector domain? Would the connector domain fit or not fit in terms of its volume/shape? The authors should consider clarifying in the manuscript the level of certainty with which they claim the extra density they observe is accounted for by an additional copy of the C-terminal domain.

In the BornaL model the connector domain is 180 residues and forms an approximately spherical domain (dimensions in x,y,z of 34-38angstrom). The BornaL CTD domain is 130 residues and approximately elliptical (dimensions of 42 angstrom along the major axis, and 23-24 angstrom along the minor axis). The extra density

we observe, though at low resolution, has elliptical features which adequately accommodate the CTD much better than the connector domain. In the comparison of CTD vs CD we conclude the CTD to be more likely. However, we are open to this density potentially corresponding to a different fragment of the L protein or a minor amount of a contaminating host protein.

We have revised the manuscript (lines 301–310) to ensure our level of confidence is accurately represented and introduces the possibility of this being a low abundance contaminating protein. The new text is below.

“Detailed analysis of both cryoEM samples revealed minor populations of particles which contained additional map density or BoDV-1 L-proteins. In the RNA free sample iterative rounds of classification from the consensus refinement revealed a set of particles, comprising approximately 2% of the total, which contained additional density between the CTD and RdRp (Supplementary Fig. 1, Supplementary Fig. 6a, b). Given this high purify of the sample and that we have been able to resolve the ordered regions from the single BoDV-1 LP complex, we tested whether single BoDV-1 L domains would fit in this density. Using the size and shape constraints of the density we concluded that the CTD was the best fit of the three domains in the C-terminal region (Supplementary Fig. 6c). Classification of these particles in a much larger box did not yield observations of full dimeric particles. This suggests a complex of BoDV-1 LP with an additional copy of the CTD can form.

Due to the low resolution of the map in this region we are unable to uniquely orient the domain in the density or exclude that this is a contaminating protein from the expression cells present at low concentration. Recent structural data on the hPIV3 L-protein showed an additional copy of the CD in the model ¹⁵ in an oligomeric assembly. However, this was located on a different surface of the RdRp domain and the hPIV3 map was determined to a much higher resolution. While an interesting observation, we are unable to conclude the biological relevance of this observation.

“

Reviewer #3 (Remarks to the Author):

The revised manuscript has addressed my concerns.

Suggestion: Move the polymerase activity data (Suppl Fig2) to main text, given its importance.

As suggested, we have moved this figure to the main text as figure 3. Main body and supplementary figures have been renumbered to reflect this.

Reviewer #4 (Remarks to the Author):
